# The Usefulness of the QR Code in Orthotic Applications after Orthopedic Surgery

**DOI:** 10.3390/healthcare9030298

**Published:** 2021-03-08

**Authors:** Jaeho Cho, Gi-Won Seo, Jeong Seok Lee, Hyung Ki Cho, Eun Myeong Kang, Jahyung Kim, Dong-Il Chun, Young Yi, Sung Hun Won

**Affiliations:** 1Department of Orthopaedic Surgery, Chuncheon Sacred Heart Hospital, Hallym University, 77, Sakju-ro, Chuncheon-si 24253, Korea; hohotoy@nate.com; 2Department of Orthopaedic Surgery, Soonchunhyang University Seoul Hospital, 59, Daesagwan-ro, Yongsan-gu, Seoul 04401, Korea; 102980@schmc.ac.kr (G.-W.S.); 124856@schmc.ac.kr (J.S.L.); 125134@schmc.ac.kr (H.K.C.); 129741@schmc.ac.kr (E.M.K.); hpsyndrome@naver.com (J.K.); orthochun@gmail.com (D.-I.C.); 3Department of Orthopaedic Surgery, Seoul Foot and Ankle Center, Inje University Seoul Paik Hospital, 85, 2-ga, Jeo-dong, Jung-gu, Seoul 04551, Korea; 20vvin@naver.com

**Keywords:** QR code, orthosis, orthopedic surgery, patient education

## Abstract

The purpose of this study is to evaluate the utility of QR (quick response) codes in explaining the proper method for orthotic use after orthopedic surgery. A questionnaire survey was adopted to evaluate patient satisfaction with education and training in orthotic applications after orthopedic surgery. The study periods were 1 April to 30 April 2017, and 1 October to 31 October 2017. The oral training involving the conventional orthoses was conducted in April, and the videos with the orthosis on the QR code were captured in October. The QR code containing the data was distributed and the education was conducted. A total of 68 patients (QR-code group: 33) participated in the questionnaire survey. After the QR code application, the number of retraining cases increased from 62.9 to 93.9% (*p*-value < 0.01). The mean scores of the four items measuring the comprehension increased from 10.97 to 14.39. The satisfaction level rose from 7.14 to 9.30, and the performance increased from 7.14 to 9.52 (*p*-value < 0.01). The QR code is expected to be a valuable method for explaining the orthotic application after orthopedic surgery, and especially when repeated explanations are needed for elderly patients.

## 1. Introduction

Interest in the improvement of the quality of medical care and patient safety is growing. The obligation to explain all the procedures performed by the medical staff is greatly emphasized. The provision of accurate information to the patients enables understanding of the medical services, providing them with opportunities to actively participate in the treatment [1]. However, with the increasing number of elderly patients in recent years, patient’s lack of understanding is expected, thereby increasing the necessity for repeated education, which is a burden not only for the patient but also for the healthcare personnel [2].

In orthopedic departments, nonsurgical treatments, intra-articular injections, splint fixation, and physical therapy are also used in addition to surgical treatment. The effort and patience required are especially high for patients wearing braces [3]. The medical personnel are required to provide the corresponding explanation to the patients and their caregivers.

With the recent advances in information technology (IT), it has become easier to access various audiovisual materials such as photographs and videos. In particular, this accessibility is facilitated by the smartphone popularization. The usefulness of online video tutorials is reported in medical fields, especially in the field of training of medical staff or students and in video descriptions for patients [4,5].

It has been reported that approximately 14% of the medical information provided orally is remembered correctly by the patients [6]. To address this challenge, it has been reported that the usage of pictography during medical explanations increased patient memory by 80% [7,8]. Sandberg et al. [9] reported that even if the information was not remembered at first, the patient recall ability increased to 67% when a clue was provided. Therefore, it is important to provide visual data and repeated interventions to ensure communication of medical information is memorized by the patient.

The QR (Quick Response) code is a matrix barcode (or two-dimensional barcode) that represents information in a black-and-white plaid pattern; these barcodes comprise vertical and horizontal information and contain up to 7089 numeric characters, or 4296 alphanumeric characters, or 2953 bytes (binary data). They can be used to encode specific URLs linking to sounds, pictures, and video information. Recently, the number of scanner applications capable of recognizing QR codes in smartphones has increased, and it is easier to obtain the desired information by simply recognizing the QR code with a smartphone rather than browsing the information online using a web-browser URL. Accordingly, this code technology is increasingly popular. A recent study reported the effectiveness of QR codes on a mobile phone in improving patient surveys [10].

Recently, the usage of QR codes in elderly patients with heart disease has led to increased medication compliance [11]. Likewise, they have the advantage of increasing medical accessibility to older age or chronic disease patients who require repetitive and continuous education.

This study evaluates the usefulness and patient satisfaction with orthopedic orthoses for which the QR code is used.

## 2. Materials and Methods

All research was performed in accordance with the relevant guidelines and regulations. The study periods were 1–30 April 2017 and 1–31 October 2017. The oral training involving conventional braces was conducted in April, and the videos with the braces on the QR code were captured in April. Therefore, we investigated two groups: the QR group as a case group and the oral training group as a control. The QR code containing the data was distributed before the education was provided. The QR code was distributed in the form of a business-card-size print and sticker, which was attached to the brace (Figure 1), while the length of each video varied from 1 to 2 min.

To evaluate the satisfaction with the training in wearing the orthosis and the degree of satisfaction associated with the orthosis after the orthopedic surgery (orthoses included shoulder-abductor brace, thoracolumbosacral orthosis (TLSO), hip-abductor brace, corset, Philadelphia cervical collar brace (P-brace)), a questionnaire was administered to patients who needed to wear it in October. The questionnaire included simple demographic data such as the patient’s gender and age, and the frequency and timing of training in orthosis wearing, along with the training contents. In order to evaluate comprehension, the questionnaire included the following items: understanding the position of the orthosis, the duration of post-discharge orthosis wearing, the precautions for orthosis wearing, and the timing of the wearing and removal of the orthosis. To evaluate patient satisfaction, the patient was asked to provide subjective scores on a scale of 1 to 10. Finally, a performance score ranging from 1 to 10 was measured by the medical staff to determine if the patient was wearing the orthosis correctly, including the wear position, fixation strength and wearing order (Figure 2). A descriptive analysis of all the variables was performed, including the mean and the standard deviation or frequency. The data normality was tested using the Kolmogorov–Smirnov test. The Student’s t-test was utilized to compare continuous variables, and the chi-square test was used to compare the categorical variables between the two groups.

## 3. Results

A total of 68 patients participated in the questionnaire survey. The mean age was 61.0 years (range: 10–88 years). The TLSO constitutes 33.8%, followed by the shoulder-abductor orthosis (32.4%), corset brace (14.7%), hip-abductor orthosis (13.2%), and P-brace (5.9%) (Table 1).

At least two-thirds of the training time in both groups was spent wearing the orthosis. The orally trained group and the group supplied with QR code were exposed to repeated training three times in 11% and 29% of the cases, respectively. Meanwhile, with the application of the QR code, the frequency of retraining (accessibility) increased from 62.9 to 93.9% (*p*-value < 0.01) (Figure 3).

The mean scores of the four items that measured the comprehension increased from 10.97 to 14.39 (*p*-value < 0.01), and the satisfaction level increased from 7.14 to 9.30. The performance increased from 7.14 to 9.52 (*p*-value < 0.01) (Figure 4).

## 4. Discussion

In this first reported study using a QR code to explain orthosis wearing to patients after orthopedic surgery, the patients who used the QR code showed sound outcomes in terms of both the wearing competency and understanding of the orthosis compared with patients who received only an oral explanation.

Yuzer et al. [12] reported that patients with stroke most often declined to wear an orthosis because they felt it was unnecessary and inconvenient. It is important for the patient to wear a brace continuously and correctly. Improper wearing of the orthosis results in discomfort, which may reduce patient compliance. There are many types of orthosis that are worn differently for varying durations. Furthermore, it is necessary for the patient to be repeatedly educated about the orthosis.

It is difficult for the patient to fit into some of the orthotic types without the help of a caregiver; therefore, the caregiver needs to have precise information. It is also difficult, however, for all caregivers to accompany the medical staff at the time of explanation, suggesting the need for a handy educational tool. Consequently, it is expected that the caregiver education will increase the compliance of patients.

As the smartphone technology becomes widespread, the QR code will be used in various industries. Because the memorizing of the URL is unnecessary and the smartphone is portable, the QR code can be used to access information conveniently and repeatedly. Accordingly, the use of the QR code for accurate drug administration in the elderly population has been reported [13]. In the orthopedic field, it has been used in cast management and sound clinical outcomes have been reported [14].

As the Internet has become widespread, various types of information can be accessed quickly and easily, but it is also difficult to identify accurate information. In the case of a nonmedical person, it is difficult to understand the correct terminology, which complicates the search and often introduces inaccuracies in medical knowledge conveyed to the patient. The usage of the QR code prevents the dissemination of false orthosis information to the patients, because it provides audiovisual information in an easy way via a smart phone. This study shows that the QR code can be used more effectively by repeated patient education and accessibility than conventional oral training in an outpatient clinic. It is expected the patient compliance will be improved by minimizing the inconvenience due to wearing of the correct orthosis and enhancing the understanding of the importance of wearing the orthosis.

Spaulding et al. described three major variables in orthotis and prosthesis care; (1) the state of functioning, disability, and health (International Classification of Functioning, Disability and Health); (2) orthotic and prosthetic technical properties, procedures, and appropriateness; and (3) professional service as part of orthotic and prosthetic interventions [15]. The usage of a QR code may enhance the provision of professional service throughout the orthotic care of patients.

Although this study is the first study demonstrating the usefulness of the QR code for orthosis application after orthopedic surgery, limitations have been identified. First, the number of the observed patients is small and the follow-up period is short. However, it is significant that the QR code, which has the advantages of repeatability and convenience, was newly used in the orthotic field. Further, the QR code is widely available in various medical fields beyond orthopedics. Second, our study compared QR codes with oral explanations about the overall understanding of orthosis after orthopedic surgery. Because this is the first attempt at orthopedic surgery, we have investigated a number of orthoses, suggesting possible heterogeneity. In future studies, each orthosis application and training will be investigated comprehensively. Third, our study did not compare online educational tutorials containing educational videos with our QR code containing an educational video. This is considered a study limitation, and it will be necessary to compare the two videos in the future. However, the advantages of our study are two-fold. First, the educational and training videos included in this study are tailored to the patient and individual circumstances. Therefore, patients can obtain a better understanding of brace application. Second, it is easier for elderly patients to access information using the QR code than to search the Internet for every educational video. Lastly, although we did not compare QR code education with a conventional paper education brochure, because of its repeatability and better visualization, QR code education can be a better option for patients.

## 5. Conclusions

The QR code is a valuable method for training patients and caregivers regarding brace applications after orthopedic surgery. It has gained popularity among other medical fields. Even though it has the limitation that there are few studies showing superiority compared to conventional education modalities, the authors believe that the QR code will be used in several medical fields in the future because of its repeatability and accessibility.

## Figures and Tables

**Figure 1 healthcare-09-00298-f001:**
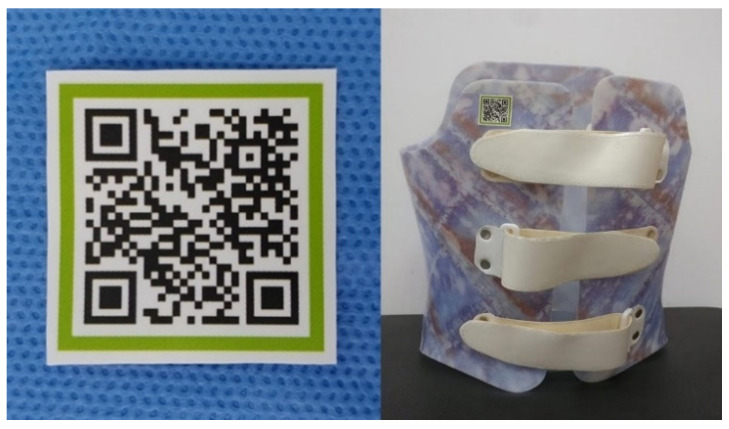
The QR code was distributed in the form of a business card-sized print and sticker and the sticker was attached to the brace.

**Figure 2 healthcare-09-00298-f002:**
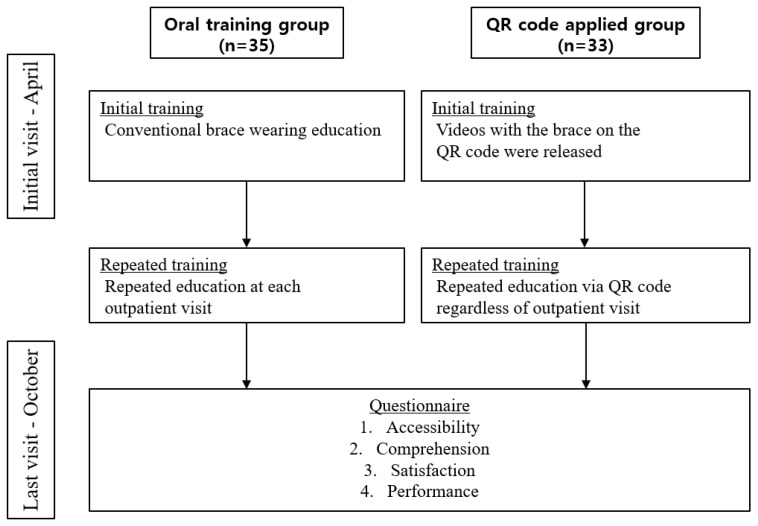
Flow chart of the study process.

**Figure 3 healthcare-09-00298-f003:**
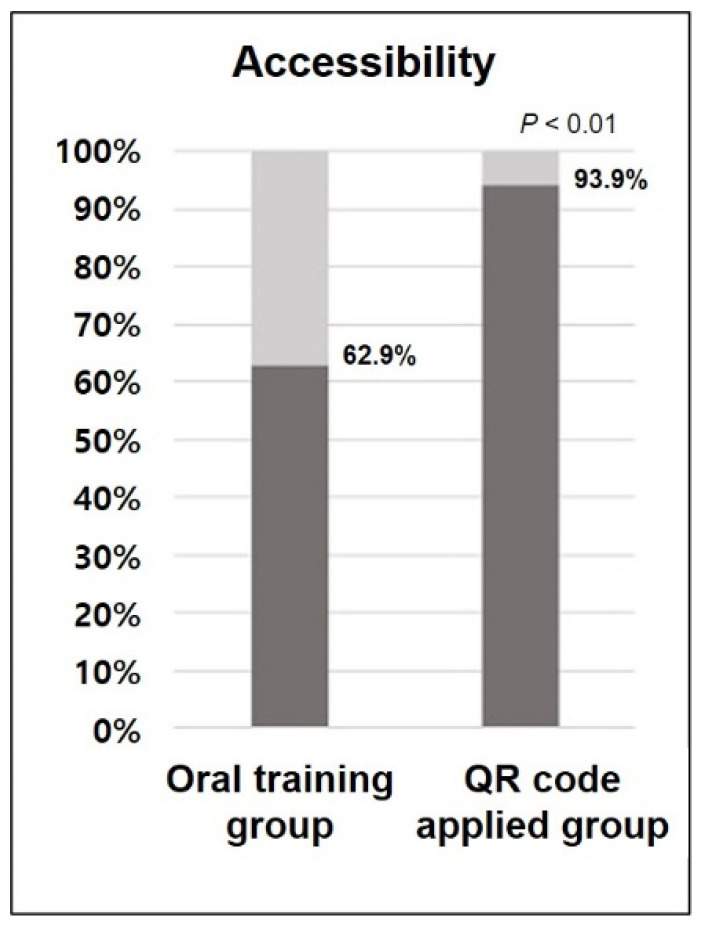
The difference in frequency of retraining (accessibility) between the oral training group and QR code applied group.

**Figure 4 healthcare-09-00298-f004:**
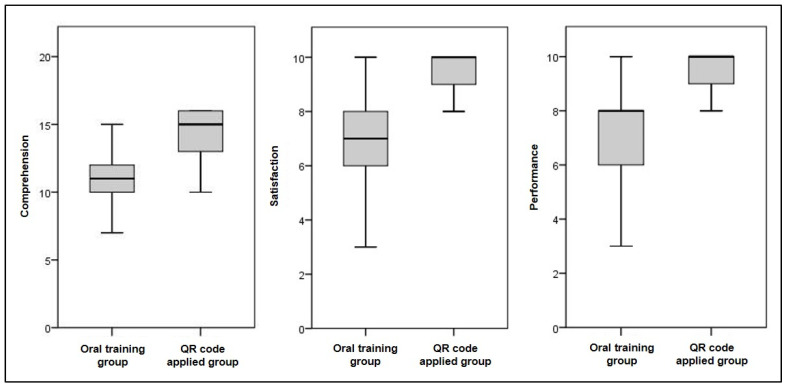
The mean scores of the four items measuring comprehension, satisfaction level, and performance score.

**Table 1 healthcare-09-00298-t001:** Patient demographics.

		Group		Total (%)	*p* Value
		Oral Training Group	QR Code Applied Group		
Age	Mean ± SD	58.60 ± 16.20	63.55 ± 15.93	61.0 ± 16.14	0.209 *
	Range (Min–Max)	10–88	15–85	10–88	
Sex	Male	20	19	39 (57.4)	0.971 †
	Female	15	14	29 (42.6)	
Orthosis	Shoulder abduction brace	15	7	22 (32.4)	0.354 †
	TLSO	11	12	23 (33.8)	
	Hip abduction brace	4	5	9 (13.2)	
	Corset	4	6	10 (14.7)	
	P-brace	1	3	4 (5.9)	

*p* values calculated by * *t*-test or † Chi-square test. SD = standard deviation; Min = minimum; Max = maximum; TLSO = thoracolumbosacral orthosis.

## Data Availability

The data presented in this study are available on request from the corresponding author. The data are not publicly available due to data restriction policies.

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
