# Peer review of "The Usefulness of the QR Code in Orthotic Applications after Orthopedic Surgery"

_healthcare, 2021, doi:10.3390/healthcare9030298_

Round 1
Reviewer 1 Report
Authors have utilized the RQ code to enhance orthotic application use and thusly improve patient satisfaction.
Although this addresses one major challenge of Patient follow up and that data acquisition has its hardships.
The methodology can be articulated in a better way and much more clear way. I flow chart is recommended. from what I could understand from the results, not the methods or the introduction that the researchers gave the patients the QR code so the can scan with their phones to watch and re-watch the instruction of use or training on how to apply their orthotic applications. While the control group was given the instructions only once by trainers in an oral form, (no visual assistance or aids?!).
Then in some way un clear to me the authors surveyed the satisfaction.
it not clear if patients where invited in for the survey, called over phone or those with the QR had digital survey and the other paper based?
all in all the study is interesting however, the motivation or rational for its conducting is not clear and needs to be concreted further.
Author Response
We really appreciate your thoughtful comment.
We authors made some modification according to your advice
First, We improved methodology part using flow chart for better understanding of our study protocol.
Second, In terms of satisfaction, it was evaluated via questionnaire and we clarified the statement on manuscript.
Third, We made enhancement in 4th paragraph and 5th paragraph of introduction part to emphasize our study purpose and rational
Again, We are really grateful to improve our manuscript by your advice.

Reviewer 2 Report
The introduction section is too short.
Moreover, a lot of applications in several domains use QR code. Therefore, the use is not novel. This is at maximum a new application but using a well known and widely used method (QR codes).
The paper should list the summaries the main novel contribution in a detailed manner to justify its novelty.
Authors need to provide more discussion about the limitation and practicality in the analytical section. QR codes have limitations please define its limitations clearly.
Authors should provide some scientific insight into the conclusion. Conclusions are just a sentence. The problem statement is missing.
The comparison with related works is missing. As there are many works attempts using QR code in the medical field. moreover, this comparison should be quantitative and focused on the technical advantages and limitations of Qr codes when compared with other technologies.
The conclusion should include the main findings and implications of the results.
Author Response
We really appreciate your thoughtful comment.
First, We authors made some modification according to your advice, so we add more backgrounds of QR code application on introduction part for more comprehension. We made enhancement in 4th paragraph and 5th paragraph of introduction part to emphasize our study purpose and rational
Second, It's true that we didn't make statement properly about limitation, so we modified the manuscript and put more explanation of limitation on 7th paragraph of discussion part(lack of comparement with conventional paper education brochure)
Lastly, in conclusion part, we clarified our message and implication of study more comprehensively
Again, We are really grateful to improve our manuscript by your advice.

Reviewer 3 Report
This manuscript should be improved in three main respects in order to be ready for publication:
- It contains some misleading assertions about the nature of QR codes. QR codes are machine-readable codes that can contain up to 7089 numeric characters, or 4296 alphanumeric characters, or 2953 bytes (binary data). They do not contain, as stated by the authors "sound, picture and data information", but rather they can be used to encode specific URLs linking to those types of information.
- The materials and methods should be better described. It is unclear whether or not all patients underwent oral training and whether the videos accessed via the QR code were recordings of the oral training or not. Nor is the exact meaning of the following statement clear: "After the application of the QR code, the frequency of retraining (accessibility) increased by ..." What, if any, were the constraints for oral training? How was the oral training performed and how could it be enjoyed by the users?
- There are some missed descriptions and references . In particular, it would useful to add some (short) descriptions and citations about both the QR code technology and the statistical tests, and their use in this work.
Author Response
We really appreciate your thoughtful comment.
We authors made some modification according to your advice
- We changed the manuscript and modified the definition of the QR code according to your comment on 4th paragraph of introduction part.
- And we put additional figure of flow chart, to clarify the difference between case and control group. The flow chart contains concrete information about how the patients being educated.
Again, We are really grateful to improve our manuscript by your advice.

Round 2
Reviewer 2 Report
A lot of applications in several domains use QR code. Therefore, the use is not novel.
The paper does not provide a clear list of contributions in a detailed manner to justify its novelty.
The authors do not provide an in-depth discussion about the limitations.
The authors do not provide scientific insights into the conclusion.
The conclusion section still very limited.
The comparison with related works is not conducted.
Consequently, I am not satisfied with the author's revision.
Author Response
We really thank you for your comment on our manuscript.
We do agree your opinion on that the usage of QR code lacks novelty.
So we revised the title and manuscript into ‘The usefulness of the QR code in orthotic applications after orthopedic surgery’, not to make misunderstanding but to circumstantiated our message.
To support our conclusions on the usefulness of the use of QR codes, we added more references and discussed, and we did our best to revise the manuscript.
Again, We really appreciate your detailed advice.

Reviewer 3 Report
The authors have provided to clarify the way in which they have conducted their test, also thanks to the aid of a diagram. Now the procedure adopted is quite clear, and the way in which it was conducted and quantified seems to be scientifically correct. However, there are still several gaps in the presentation of the manuscript that prevent it from being published in its current form.
First of all, what has been described about QR-codes in several parts of the manuscript still suggests that this technology represents in itself something special for the application domain considered. However, the added value derives not so much from using QR codes or other types of coding techniques for URLs, but rather the possibility of being able to access explanatory audiovisuals in an easy way for patients. In this regard it is important to point out that the only novelty of this study concerns the comparison of oral learning on the use of orthopedic orthosis through scheduled meetings with medical staff versus learning on this subject conveyed through short audio-visuals that can be consulted at will by patients thanks to the use of their smartphone. This is the real contribution of the work and authors should rephrase the title, abstract, introduction and conclusions to take this into account. The title of the manuscript itself seems to suggest that there is a novelty in which QR-codes are used in the present work, when instead they are used as usual to point to resources on the Internet.
Second, most of the contents now described in the "Discussion" section must be reorganized and inserted in a special "Related works" section immediately after the introduction.
Finally, there are some sentences that are rather inaccurate and need to be corrected in order to improve the quality of the presentation; I quote here the ones that seem most relevant to me:
line 36: However, with the increasing number of elderly patients in recent years, the procedures provided by the medical staff are unclear, ... [explain better]
line 50: ..is a matrix of the two-dimensional barcodes that represent information in a black-and-white plaid pattern..["a matrix of the two-dimensional barcodes" is incorrect]
line 61: Likewise, old ages or chronic disease patients who requires repetitive and continuous education, this suggests the advantage of increasing medical accessibility. [express better]
line 64: This study evaluates the usefulness and patient satisfaction with orthopedic orthosis for which the QR code is used ["for wich the QR code is used" sounds inadequate to describe the work]
line 127: After the application of the QR code, ... [the use of "after" is inappropriate]
line 131: The frequency of retraining (accessibility) after the application of the QR code. [see above]
line : Lastly, although we did not compare QR code education with conventional paper education brochure, but due to it’s repeatability and visualization QR code education can be better option to patients. [circumstantiate and express better]
Author Response
Reviewer 3
The authors have provided to clarify the way in which they have conducted their test, also thanks to the aid of a diagram. Now the procedure adopted is quite clear, and the way in which it was conducted and quantified seems to be scientifically correct. However, there are still several gaps in the presentation of the manuscript that prevent it from being published in its current form.
Answer:
We authors are really grateful for your thoughtful comment. And made a few more revisions.
First of all, what has been described about QR-codes in several parts of the manuscript still suggests that this technology represents in itself something special for the application domain considered. However, the added value derives not so much from using QR codes or other types of coding techniques for URLs, but rather the possibility of being able to access explanatory audiovisuals in an easy way for patients. In this regard it is important to point out that the only novelty of this study concerns the comparison of oral learning on the use of orthopedic orthosis through scheduled meetings with medical staff versus learning on this subject conveyed through short audio-visuals that can be consulted at will by patients thanks to the use of their smartphone. This is the real contribution of the work and authors should rephrase the title, abstract, introduction and conclusions to take this into account. The title of the manuscript itself seems to suggest that there is a novelty in which QR-codes are used in the present work, when instead they are used as usual to point to resources on the Internet.
Answer:
First, We totally agree your opinion on that the usage of QR code lacks novelty. So we revised the title and manuscript into ‘The usefulness of the QR code in orthotic applications after orthopedic surgery’, not to make misunderstanding but to circumstantiated our message. And added few more statements to emphasize that usage of QR code has advantage on provision of audiovisual information in an easy way.
Second, most of the contents now described in the "Discussion" section must be reorganized and inserted in a special "Related works" section immediately after the introduction.
Answer:
Second, we do agree that some of our initial mansucript in Discussion sections contains information about related works so we moved that part into introduction section
Finally, there are some sentences that are rather inaccurate and need to be corrected in order to improve the quality of the presentation; I quote here the ones that seem most relevant to me:
Answer:
Third, we appreciate your delicate correction on our manuscript so we revised the sentences.
line 36: However, with the increasing number of elderly patients in recent years, the procedures provided by the medical staff are unclear, ... [explain better]
- Thank you, We clarified the text.
line 50: ..is a matrix of the two-dimensional barcodes that represent information in a black-and-white plaid pattern..["a matrix of the two-dimensional barcodes" is incorrect]
- Thank you, We made revision on the text
line 61: Likewise, old ages or chronic disease patients who requires repetitive and continuous education, this suggests the advantage of increasing medical accessibility. [express better]
- Thank you, We clarified the text.
line 64: This study evaluates the usefulness and patient satisfaction with orthopedic orthosis for which the QR code is used ["for wich the QR code is used" sounds inadequate to describe the work]
- Thank you, We clarified the text.
line 127: After the application of the QR code, ... [the use of "after" is inappropriate]
- Thank you, We made revision on the text
line 131: The frequency of retraining (accessibility) after the application of the QR code. [see above]
- Thank you, We clarified the figure legend.
line : Lastly, although we did not compare QR code education with conventional paper education brochure, but due to it’s repeatability and visualization QR code education can be better option to patients. [circumstantiate and express better]
- Thank you, We clarified the text.
Again, We really appreciate your detailed advice.
